# Explainable AI for Retinoblastoma Diagnosis: Interpreting Deep Learning Models with LIME and SHAP

**DOI:** 10.3390/diagnostics13111932

**Published:** 2023-06-01

**Authors:** Bader Aldughayfiq, Farzeen Ashfaq, N. Z. Jhanjhi, Mamoona Humayun

**Affiliations:** 1Department of Information Systems, College of Computer and Information Sciences, Jouf University, Sakaka 72388, Saudi Arabia; bmaldughayfiq@ju.edu.sa; 2School of Computer Science, SCS, Taylor’s University, Subang Jaya 47500, Malaysia; farzeen.ashfaq@sd.taylors.edu.my (F.A.); noorzaman.jhanjhi@taylors.edu.my (N.Z.J.)

**Keywords:** retinoblastoma, explainable AI, deep learning, LIME, SHAP, medical image analysis, InceptionV3, transfer learning

## Abstract

Retinoblastoma is a rare and aggressive form of childhood eye cancer that requires prompt diagnosis and treatment to prevent vision loss and even death. Deep learning models have shown promising results in detecting retinoblastoma from fundus images, but their decision-making process is often considered a “black box” that lacks transparency and interpretability. In this project, we explore the use of LIME and SHAP, two popular explainable AI techniques, to generate local and global explanations for a deep learning model based on InceptionV3 architecture trained on retinoblastoma and non-retinoblastoma fundus images. We collected and labeled a dataset of 400 retinoblastoma and 400 non-retinoblastoma images, split it into training, validation, and test sets, and trained the model using transfer learning from the pre-trained InceptionV3 model. We then applied LIME and SHAP to generate explanations for the model’s predictions on the validation and test sets. Our results demonstrate that LIME and SHAP can effectively identify the regions and features in the input images that contribute the most to the model’s predictions, providing valuable insights into the decision-making process of the deep learning model. In addition, the use of InceptionV3 architecture with spatial attention mechanism achieved high accuracy of 97% on the test set, indicating the potential of combining deep learning and explainable AI for improving retinoblastoma diagnosis and treatment.

## 1. Introduction

Retinoblastoma is a rare and aggressive eye cancer that affects infants and young children, with an incidence rate of about 1 in 15,000 live births worldwide [1]. Figure 1 shows the lateral view of a normal eye compared to an eye having retinoblastoma.

Early detection and treatment of retinoblastoma is crucial for preserving vision and preventing metastasis, but it remains challenging due to the complex and diverse clinical features of the disease [3]. Fundus photography is a non-invasive and widely used imaging technique for diagnosing and monitoring retinoblastoma, which involves capturing images of the retina using a specialized camera and analyzing them for abnormalities [4]. However, accurate and reliable interpretation of fundus images requires extensive training and expertise, which is often limited in resource-constrained settings [5]. The International Intraocular Retinoblastoma Classification (IIRC) is a standardized system for describing the clinical features of retinoblastoma based on fundus imaging [6]. Figure 2 shows an example of the IIRC classification system, which can aid in the diagnosis and management of retinoblastoma.

Deep learning models have shown remarkable progress in every domain [7,8,9,10], particularly in medical image analysis, including retinoblastoma detection from fundus images [11,12,13,14]. These models learn to automatically extract relevant features and patterns from the input images and use them to make predictions with high accuracy and speed [15]. However, their decision-making process is often considered a “black box” that lacks transparency and interpretability, which can hinder their clinical adoption and trust [16,17,18]. To address this challenge, several explainable AI techniques have been developed to generate local and global explanations for the predictions of deep learning models, including LIME (local interpretable model-agnostic explanations) [19] and SHAP (Shapley additive explanations) [20]. These techniques aim to identify the regions and features in the input images that contribute the most to the model’s predictions and provide insights into the decision-making process of the models [21,22,23,24,25]. In this project, we explore the use of LIME and SHAP to generate explanations for a deep learning model trained on retinoblastoma and non-retinoblastoma fundus images. We collected and labeled a dataset of 400 retinoblastoma [26] and 400 normal fundus images extracted from [27], split it into training, validation, and test sets, and trained a deep learning model using transfer learning from a pre-trained InceptionV3 architecture [28]. We then applied LIME and SHAP to generate explanations for the model’s predictions on the validation and test sets, and evaluated their effectiveness in diagnosing and differentiating retinoblastoma from non-retinoblastoma. Our study contributes to the growing body of literature on using explainable AI for improving medical image analysis and diagnosis and provides insights into the interpretability and transparency of deep learning models for retinoblastoma detection.

Hence, we can say that despite the recent advances in deep-learning-based medical image analysis, the lack of interpretability of these models hinders their application in clinical practice. This is especially important in the context of retinoblastoma. While deep learning models have achieved remarkable performance in detecting retinoblastoma, their black-box nature poses significant challenges in explaining the decision-making process. There is a clear gap in the literature when it comes to the application of explainable AI techniques in retinoblastoma detection. According to a recent review of the literature by [29], interpretability is a key issue in the application of deep learning to medical image analysis, and explainable AI techniques are becoming increasingly important in this field. In the specific context of retinoblastoma detection, we propose an explainable AI system that combines a deep learning model with a saliency map to identify the important regions in retinal images for diagnosis. Our main contributions are:We propose a novel approach to improve the interpretability of deep learning models for retinoblastoma detection.We apply LIME and SHAP, two popular explainable AI techniques, to generate saliency maps and identify the most important features contributing to the model’s decision.To the best of our knowledge, this is the first study to apply LIME and SHAP for the task of retinoblastoma detection.Our proposed approach provides insights into the decision-making process of the model and enables clinicians to better understand and trust the model’s predictions.Our work contributes to the growing body of literature on the application of explainable AI in medical image analysis, specifically in the context of retinoblastoma detection.

The remainder of the paper is structured as follows: Section 2 discusses the related work section, including the most recent and relevant studies on deep-learning-based retinoblastoma detection and explainable AI. Then in Section 3, we describe the dataset and preprocessing steps in the data section. Additionally, in the same section we discuss the main methods, including LIME and SHAP, and how we use them to improve the interpretability of deep learning models for retinoblastoma detection. Section 3.4 discusses the experimental setup for the training process and Section 3.5 discusses the training hyperparameters. Section 4 outlines the results of both classification and explainable visualizations. Section 5 presents a brief discussion on the results compared with benchmarks. Section 6 covers the potential constraints and obstacles that must be recognized. Finally, in Section 7, we summarize the main contributions of our article, identify the strengths and weaknesses of our approach, and recommend future research directions.

## 2. Literature Review

Retinoblastoma is a rare and potentially fatal type of pediatric cancer that affects the retina of the eye. Early detection and treatment are crucial for preventing vision loss and saving lives. In the clinical context of retinoblastoma diagnosis, various approaches are employed worldwide, including routine screening and opportunistic screening.

For the early detection of retinoblastoma, organized screening programs are in existence in various nations. These programs often involve routine eye exams for newborns and young children to look for any disease-related symptoms. For instance, some nations have put in place governmental initiatives that encourage routine newborn eye screenings and subsequent follow-up exams to find retinoblastoma in its earliest stages [30]. These screening methods are designed to find retinoblastoma before symptoms appear, allowing for prompt treatment and intervention [31].

In contrast, other countries rely on opportunistic screening, wherein eye examinations are conducted when an individual presents with symptoms or risk factors associated with retinoblastoma [32]. This approach often depends on the awareness and proactive behavior of parents, caregivers, and healthcare professionals in recognizing potential signs of retinoblastoma, such as leukocoria (white pupil reflex) [33] or strabismus (misaligned eyes) [34]. While opportunistic screening may help detect some cases of retinoblastoma, it can be limited by the variability in symptom presentation and the reliance on individual awareness [35].

However, there are limitations to the accuracy of conventional screening methods and opportunistic screening systems, particularly when it comes to spotting small or deep-seated tumors. Ophthalmoscopy and ultrasonography are the major traditional procedures for finding these tumors, and while they have historically been highly successful, they do have certain drawbacks.

Ophthalmoscopy is a visual inspection of the retina that can detect abnormalities such as tumors, but it requires a highly trained and experienced clinician to perform the examination accurately. Moreover, it can only visualize the outer layer of the retina, which may not always reveal the presence of a tumor in its early stages [36,37].

Ultrasonography is another technique that is often used to diagnose retinoblastoma. It involves using high-frequency sound waves to create images of the eye. However, it also has limitations, as it cannot always distinguish between benign and malignant tumors, and it is unable to detect small tumors that are located in certain parts of the eye [38].

In general, conventional approaches have been somewhat successful in detecting retinoblastoma [39,40,41], but they are challenging due to their reliance on the clinician’s skill and their inability to pick up on profound or miniscule tumors [42,43]. This emphasizes the requirement for more sophisticated, automated and trustworthy diagnostic methods, such as deep learning algorithms, to enhance the efficacy and accuracy of retinoblastoma identification.

In the clinical setting of retinoblastoma diagnosis, AI can provide valuable support by enhancing accuracy, efficiency, and accessibility. AI systems can examine retinal scans and help doctors identify probable retinoblastoma symptoms, facilitating an early diagnosis and prompt treatment. For example:In regions where access to specialized ophthalmologists is limited, AI-powered retinoblastoma screening systems can be deployed. Through non-invasive methods like smartphone-based fundus photography, these devices can automatically analyze retinal images. The retinoblastoma signs can be promptly and precisely identified by the AI algorithms, alerting medical professionals to possible cases that need additional investigation.AI can assist clinicians in distinguishing between benign and malignant retinal tumors. This capacity is especially useful in situations where differentiation is difficult based just on visual assessment. By analyzing various features within the retinal images, AI models can provide additional diagnostic information, aiding in appropriate treatment planning and reducing unnecessary invasive procedures.AI can also help medical personnel track the development of retinoblastoma and evaluate therapy effectiveness. By comparing sequential retinal images over time, AI algorithms can detect subtle changes in tumor size, shape, or characteristics, enabling early detection of tumor growth or recurrence. This timely information empowers clinicians to adjust treatment strategies promptly and optimize patient outcomes.

Deep learning algorithms, a subfield of AI, have emerged as a well-known method for the detection and diagnosis of several ocular fundus diseases, including diabetic retinopathy [44,45], glaucoma [46,47], and age-related macular degeneration [48,49]. Large datasets of annotated photos are used to train these algorithms so they can recognize the distinctive traits and patterns linked to certain diseases. Furthermore, deep learning algorithms have also recently demonstrated impressive performance in diagnosing retinoblastoma from fundus images with high accuracy [50,51,52,53]. These algorithms use artificial neural networks to learn and identify specific features and patterns in fundus images that are associated with retinoblastoma. So, compared to conventional methods, automated deep-learning-based medical image analysis has a number of benefits, including increased accuracy, scalability, and speed.

The study by Kaliki et al. [52] explores the application of AI and ML in the diagnosis and management of retinoblastoma. Their qualitative investigation included 109 eyes with 771 fundus pictures, 590 of which showed iRB. The trained AI model’s sensitivity, specificity, positive predictive value, and negative predictive value were, respectively, 85%, 99%, 99.6%, and 67%. Their results show promising accuracy rates and suggest that AI has the potential to improve the diagnosis and treatment of retinoblastoma. Kumar [54] proposed a CNN-based method for detecting retinoblastoma. The proposed method utilized an automated thresholding method to identify the tumor-like region (TLR) in retinoblastoma, followed by the use of ResNet and AlexNet algorithms for classification of cancerous regions. The performance of ResNet and AlexNet was compared, and ResNet50 was found to yield the highest classification performance of 93.16%. In another paper, by Jebaseeli et al. [55], the focus is on the detection of retinoblastoma through image analysis. In their proposed work, the fundus picture is preprocessed to eliminate noise brought on by light during eye scanning or recording using a linear predictive decision-based median filter. After that, a 2.75D CNN approach is used to segment the images in order to separate the foreground tumor cells from the backdrop. On the basis of the tumor stage of malignancy, the tumors are then divided into additional categories. The accuracy, sensitivity, and specificity of the suggested method are greatly increased to 98.84%, 97.96%, and 98.32%, respectively. The study conducted by Rahdar et al. [56] focused on developing a semi-supervised segmentation method for retinoblastoma tumors in fundus images. The proposed method utilized a combination of unsupervised clustering and supervised classification techniques to extract the tumor regions from the fundus images. The authors evaluated the performance of their method on a dataset of 80 fundus images and compared it with other state-of-the-art methods. The GMM was selected as the clustering algorithm due to its capability of drawing elliptical boundaries and its probabilistic nature. Overall, the study presented a promising approach for the accurate and efficient segmentation of retinoblastoma tumors in fundus images, which could have potential implications for the diagnosis and treatment of retinoblastoma.

In another study, conducted by [57], the authors describe a multi-view convolutional neural network for automated ocular anatomy and tumor segmentation in retinoblastoma patients using MRI. The researchers used intra-class correlation and the dice similarity coefficient to assess morphological and spatial performance in 40 retinoblastoma and 20 healthy eyes from 30 individuals. The best results were obtained by using a three-level pyramid MV-CNN and data enhancement. The findings demonstrated that MV-CNN can accurately segment retinoblastoma ocular structures and tumors. Henning et al. [58] discuss the use of CNNs to detect leukocoria, or white-eye reflections, which is a prominent symptom of retinoblastoma. The researchers trained several CNNs of varying architectures and depths. The researchers found that CNNs produced much better results than traditional fully-connected neural networks and that biased architecture of CNNs allows for a fully trainable system without requiring hand-coded feature extractors. The researchers also found that a small capacity network achieved better results for their task than larger capacity networks. In another work, by [59], a deep Visual Geometry Group-net CNN classifier is proposed for the automatic detection of leukocoria. The study takes use of a database of 124 eye scans, 35 of which are classified as ’leukocoric’ and the remaining as ’healthy’. Moreover, the study looks into the usage of deep features extracted from the VGG-Net for leukocoria identification tasks.

However, all of these studies mainly focus on detection and segmentation. Zhang et al. [60] developed the Deep Learning Assistant for Retinoblastoma (DLA-RB), which uses uses explainable AI to generate visualizations by Grad-CAM to highlight the regions of an image that are most important in the algorithm’s prediction. Statistical analysis was conducted using both R-Statistical Software (version 4.1.1, R Foundation for Statistical Computing, Vienna, Austria) and Stata (version 17.0, StataCorp LLC, College Station, TX, USA). The efficiency of the algorithm was compared to that of human ophthalmologists, and a cost-utility study revealed that their proposed system is more affordable for both retinoblastoma identification and tumor activity surveillance.

Despite these promising results, there is still a lack of studies that have applied explainable AI techniques to retinoblastoma detection. Most studies to date have focused on the development of deep learning models and have not explored the potential of explainable AI to improve their interpretability and trustworthiness. This research gap highlights the need for further studies that explore the use of explainable AI in retinoblastoma detection and its potential impact on clinical decision-making.

In this paper, we present a novel approach for retinoblastoma detection that combines deep learning models with explainable AI techniques. We use a dataset of 800 fundus images, half of which are labeled with retinoblastoma and the other half are non-retinoblastoma. We first train a deep learning model to classify the images into retinoblastoma and non-retinoblastoma categories and then apply LIME and SHAP to generate visual explanations for the model’s predictions. Our contributions include the development of a novel retinoblastoma detection approach that combines deep learning models with explainable AI techniques, and an evaluation of the effectiveness of LIME and SHAP in improving the interpretability and trustworthiness of the model.

## 3. Materials and Methods

In this section, we provide an overview of the materials and methods used in our study. We describe the dataset collection process, preprocessing steps, model architecture, training procedure, and evaluation metrics employed. Figure 3 provides an illustrative representation of the overall framework, depicting the sequential steps involved in the data collection, preprocessing, model training, and the application of explainable AI techniques (LIME and SHAP) for generating local and global explanations.

### 3.1. Gathering Data and Preprocessing

We collected a dataset of retinoblastoma and non-retinoblastoma fundus images from multiple sources, including the MathWorks Retinoblastoma Dataset [26] and Google Images. The MathWorks dataset contained 140 retinoblastoma images, while the additional retinoblastoma images were obtained from Google Images. We ensured accurate labeling of the retinoblastoma images by consulting with two expert ophthalmologists.

We extracted non-retinoblastoma images from the Messidor dataset [27] by filtering for grade level 0 (no visible lesion) and resized all images to 224 × 224 pixels. We randomly split the combined dataset of retinoblastoma and non-retinoblastoma images into training (60%), validation (20%), and test (20%) sets using stratified sampling to ensure a balanced distribution of images in each set.

The images were preprocessed by normalizing the pixel values to the range [0, 1] and applying data augmentation techniques to the training set, including rotation, zooming, and flipping. Figure 4 shows some sample images from our dataset.

### 3.2. Deep Learning Model

Deep learning has become increasingly popular in the medical field for image classification tasks, including the detection of retinoblastoma, a rare and aggressive form of childhood eye cancer. In this study, we developed a deep learning model based on transfer learning and the InceptionV3 architecture to accurately classify retinoblastoma from fundus images. Figure 5 shows the overall flow diagram of our model.

#### 3.2.1. Transfer Learning

Transfer learning has been widely used in deep learning applications, including medical image analysis, due to the limited availability of large annotated datasets. In our study, we used transfer learning to overcome the challenge of a small dataset by leveraging a pre-trained deep learning model and fine-tuning it for our specific task of retinoblastoma classification.

#### 3.2.2. InceptionV3

InceptionV3 is a popular deep learning architecture that has been shown to achieve high accuracy in various image classification tasks. We chose InceptionV3 as the base architecture for our retinoblastoma classification model due to its high performance on similar tasks and its ability to extract meaningful features from input images. Additionally, we added two fully connected dense layers with L2 regularization and a final dense layer with sigmoid activation to the InceptionV3 architecture to improve its performance on our specific task.

#### 3.2.3. Model Architecture

We added a GlobalAveragePooling2D layer to reduce the spatial dimensions of the output from the InceptionV3 model and then added two fully connected dense layers with L2 regularization to prevent overfitting. We used an activation function of ReLU in the dense layers, which has been shown to be effective in deep learning models. To further prevent overfitting, we added dropout layers with a rate of 0.3 after each dense layer. Finally, we added a final dense layer with sigmoid activation for binary classification, as our task was to classify fundus images as either retinoblastoma or non-retinoblastoma.

### 3.3. Explainable AI Techniques

We used two popular explainable AI techniques, LIME and SHAP, to generate local and global explanations for the deep learning model’s predictions on the validation and test sets. LIME generates an interpretable model by training a local linear model around the prediction point, while SHAP provides a unified framework for feature importance estimation.

#### 3.3.1. LIME

LIME is a well-known model-independent technique for generating explanations for individual predictions generated by a black-box model. It creates a local linear model around the prediction point and weights the input features to estimate their importance in the prediction. We used the Lime package in Python to generate explanations for our model’s predictions on the validation and test sets.

#### 3.3.2. SHAP

SHAP is a game-theory technique to explain machine learning model output. It provides a unified framework for feature importance estimation and generates global explanations for the model’s behavior. We used the SHAP library in Python to generate feature importance values for our model’s predictions on the validation and test sets.

### 3.4. Experimental Setup

All experiments were conducted using Google Colab, a cloud-based platform for machine learning that provides access to powerful GPUs. We used a Tesla T4 GPU for training our deep learning model and generating explanations using LIME and SHAP. The implementation of the deep learning model and explainable AI techniques was performed in Python using the Keras and Lime packages for LIME and the SHAP library for SHAP.

### 3.5. Training Hyperparameter

Through a methodical grid search, we carefully adjusted numerous hyperparameters, including the learning rate, batch size, and number of epochs, to optimize the performance of our deep learning model. Additionally, we used a binary cross-entropy loss function and the Adam optimizer to train our model. The final architecture of our model consisted of a pre-trained InceptionV3 base with two dense layers and a final sigmoid layer for binary classification. We also used L2 regularization and dropout to prevent overfitting and improve generalization. Table 1 shows the selected hyperparameter values for model optimization.

## 4. Results

### 4.1. Results of Classification of Retinoblastoma and Normal Fundus

After training the model for 100 epochs, we achieved a training accuracy of 99% and a validation accuracy of 99%. The model also performed well on the test data, with an accuracy of 97%. The loss also decreased during the epochs, indicating that the model was learning from the data. To visualize the training and validation accuracy and loss during the training process, we plotted two separate figures. Figure 6 shows the training and validation accuracy curves, which indicates that the accuracy steadily improved during the training process for both the training and validation data. Figure 7 shows the training and validation loss curves, which indicates that the loss decreased during the training process for both the training and validation data. These results demonstrate that our deep learning model was able to accurately classify retinoblastoma and normal fundus images with high accuracy. Below, we further discuss the evaluation metrics in general with our test results on each metric.

### 4.2. Evaluation Metrics

#### 4.2.1. Accuracy

The model’s accuracy on the test set was 97%. The ratio of correct predictions to the total number of predictions made by the model is defined as accuracy. The accuracy formula is as follows:(1)$$Acc=\left(T_{P}+T_{N}\right)/\left(T_{P}+T_{N}+F_{P}+F_{N}\right)$$
where $T_{P}=$ (true positive), $T_{N}=$ (true negative), $F_{P}=$ (false positive), and $F_{N}=$ (false negative).

#### 4.2.2. Precision

The model’s precision on the test set was 98.8%. Precision is defined as the proportion of correctly predicted occurrences to the total number of positive predictions provided by the model. The following is the precision formula:(2)$$Precision=T_{P}/\left(T_{P}+F_{P}\right)$$

#### 4.2.3. Recall

The recall of the model on the test set was 99.6%. The ratio of accurately anticipated positive cases to the total number of actual positive instances in the dataset is defined as recall. The formula is as follows:(3)$$Recall=T_{P}/\left(T_{P}+F_{N}\right)$$

#### 4.2.4. F1 Score

The F1 score of the model on the test set was 99.2%. The F1 score is a harmonic average of precision and recall, and it creates a balance between the two metrics. The formula is as follows:(4)F1Score=2∗(Precision∗Recall)/(Precision+Recall)These evaluation metrics indicate that our deep learning model was able to accurately classify retinoblastoma and normal fundus images with high accuracy, precision, recall, and F1 score.

### 4.3. Visualization Results using LIME and SHAP

We used two popular explainable AI techniques, LIME and SHAP, to generate local and global explanations for the deep learning model’s predictions on the validation and test sets. LIME produced segmentations of the images and highlighted the important regions for classification. The important features of retinoblastoma fundus include yellow-white mass, calcification, retinal detachment, vitreous seeding, and subretinal fluid. LIME identified most of these features in the retinoblastoma images, but some of the segmentations were not accurate, including the outer region of the fundus. Nonetheless, the model’s important regions were identified in most of the images. Figure 8 and Figure 9 show the saliency map for two sample retinoblastoma images using SHAP.

On the other hand, SHAP provided a more accurate explanation of the model’s predictions by assigning feature importance scores to individual pixels in the image. We observed that SHAP was more effective in identifying important regions of the image, with pink areas highlighting the areas correctly identified as significant in retinoblastoma images and blue areas indicating the lack of significant features in normal images. SHAP also showed that the most important features for classification were the presence of a yellow-white mass and calcification, consistent with clinical observations. We created visualizations of the SHAP values to show the areas of the image that contributed most to the model’s prediction. These visualizations provide valuable insight into the model’s decision-making process and can aid medical professionals in interpreting the model’s predictions. Figure 10, Figure 11 and Figure 12 show the saliency map for two sample retinoblastoma images using SHAP. On the other hand, Figure 13 and Figure 14 show the SHAP values for the normal fundus images from the datset.

As can be seen from Figure 10, Figure 11, Figure 12, Figure 13 and Figure 14, in the case of retinoblastoma images, the presence of a yellow or white mass may be considered as an important feature by the SHAP algorithm, leading to the corresponding area being highlighted in pink. On the other hand, normal images may not have any significant features that impact the model output, leading to the SHAP values being blue across the entire image.

## 5. Discussion

The comparative analysis of various studies in Table 2 reveals that our study is highly competitive in terms of performance metrics such as accuracy, precision, recall, and F1 score. None of the studies in the table, except Zhang et al. [60], utilized any XAI techniques. On the other hand, our study used two popular XAI techniques, LIME and SHAP, for the interpretation of the classification model. SHAP provided a better color-coded interpretation of feature importance, which helped in identifying the most discriminative features between retinoblastoma and normal fundus images.

Comparing our study with the benchmark studies presented in Table 2, it is evident that our study outperforms most of them in terms of accuracy and other performance metrics. For instance, in the study by Kaliki et al. [52], the AI model achieved a sensitivity of 85%, specificity of 99%, PPV of 99.6%, and NPV of 67%. While their model achieved high specificity and PPV, it performed poorly in terms of sensitivity and NPV. In contrast, our model achieved a high recall of 99.6%, which indicates the ability to correctly identify positive samples. This is particularly important in medical applications where false negatives can be detrimental to patient outcomes. Similarly, in the study by Kumar [54], the CNN-based ResNet50 achieved an accuracy of 93.16%. This is significantly lower than the accuracy achieved by our model, which is 97%. This suggests that the transfer learning approach using InceptionV3 has been effective in improving the accuracy of the model. Moreover, in the study by Jebaseeli et al. [55], the linear predictive decision-based median filter and 2.75D CNN achieved an accuracy of 98.84%, sensitivity of 97.96%, and specificity of 98.32%. While their model achieved high accuracy and sensitivity, it performed slightly lower in terms of specificity compared to our model. This indicates that our model can correctly identify negative samples with high precision. Furthermore, our study used a relatively larger dataset than most of the studies in the table, which provides a more robust evaluation of the model’s performance.

## 6. Limitations

This study acknowledges some important limitations. Firstly, the potential bias arising from the dataset composition, which primarily consists of Google Images and textbook cases. Textbook cases often represent clear and well-defined instances of retinoblastoma, which may not fully capture the complexity and variability encountered in clinical practice. Real-world clinical situations might make it difficult to tell retinoblastoma from other disorders since certain cases may resemble retinoblastoma but be associated with other pathologies, or vice versa. As a result, even though the dataset used in this study provided a useful basis for training the AI model, it is crucial to recognize the potential drawbacks in extrapolating the findings to the complexity of actual clinical settings. Future investigations should aim to incorporate diverse and representative datasets that encompass a broader spectrum of retinoblastoma cases encountered in clinical practice, including those with atypical or ambiguous presentations.

Second, the absence of data on the model’s performance in actual clinical situations poses a restriction to our work. While the model has been developed and trained on a specific dataset, its effectiveness and reliability in practical healthcare applications remain unknown. The performance of the model may be impacted by a number of variables introduced by real-world scenarios, such as variations in image quality, patient demographics, and clinical circumstances. Additionally, elements absent from the training dataset, such as underrepresented rare or unusual occurrences, may have an impact on the model’s performance. It is important to acknowledge that performance observed in controlled experimental settings may not directly translate to real-world clinical practice. Thus, further research and evaluation are necessary to assess the model’s performance in diverse clinical settings and validate its usefulness as an effective diagnostic tool for retinoblastoma.

Finally, the lack of cheap and widely accessible techniques for capturing images of the eye fundus in children is also a limitation. In many healthcare settings, especially in places with low resources, traditional procedures such as ophthalmoscopy and ultrasonography may not be readily available since they frequently call for specialized equipment and trained personnel. However, the integration of AI technology in retinoblastoma diagnosis holds the promise of overcoming this limitation. With the advancements in mobile and portable imaging devices, coupled with AI algorithms, it is possible to develop cost-effective and user-friendly solutions for capturing high-quality retinal images in pediatric populations. AI algorithms can then be applied to these images to assist in the detection and diagnosis of retinoblastoma, even in settings with limited resources. Therefore, future research and development should focus on harnessing the potential of AI to address the lack of cheap techniques for capturing fundus images in children, thus enabling broader access to retinoblastoma diagnosis and improving outcomes for affected individuals.

## 7. Conclusions

In conclusion, our investigation reveals the efficiency of deep learning models in the classification of retinoblastoma and normal fundus images. Our model achieved high accuracy, recall, precision, and F1 scores on the test-set. The LIME and SHAP visualizations provided local and global explanations for the model’s predictions, highlighting important regions for classification.

One potential limitation of our study could be the size of the dataset. Although we used a comprehensive dataset for our study, it is important to note that collecting more images on a real-time basis could enhance the generalizability of our findings to larger and more diverse datasets.

Moving forward, we see two potential future directions for this work. Firstly, we can test the model’s applicability to additional ocular illnesses such as diabetic retinopathy and age-related macular degeneration. Secondly, we can investigate the use of ensemble models and model interpretability techniques such as attention mechanisms to improve the robustness and explainability of our model.

Overall, our study provides a promising avenue for the use of deep learning models in the automated diagnosis of retinoblastoma, and we hope that our findings will inspire further research in this field.

## Figures and Tables

**Figure 1 diagnostics-13-01932-f001:**
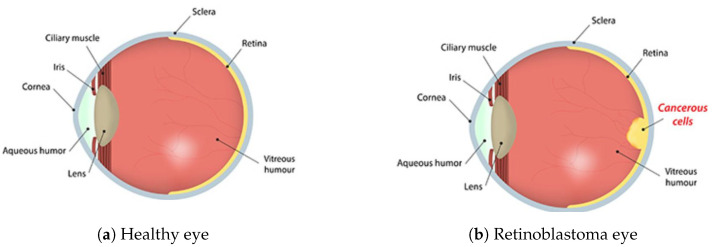
Comparison of a healthy eye (**a**) and a retinoblastoma eye (**b**) [2].

**Figure 2 diagnostics-13-01932-f002:**
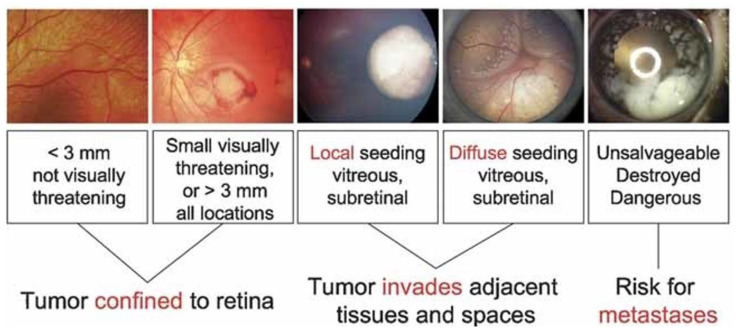
Internation Intraocular Retinoblastoma Classification [6].

**Figure 3 diagnostics-13-01932-f003:**
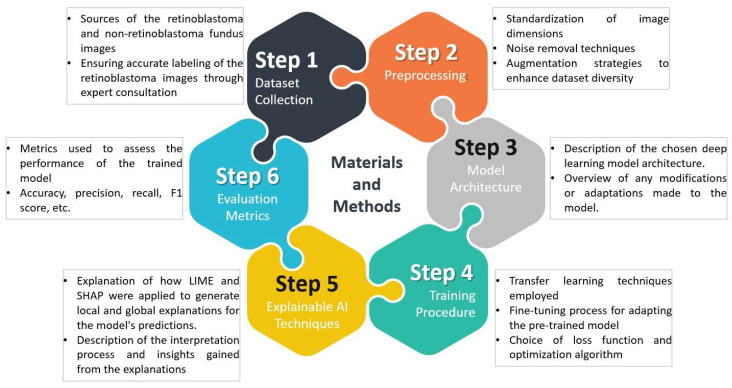
Illustration of the Overall Framework.

**Figure 4 diagnostics-13-01932-f004:**
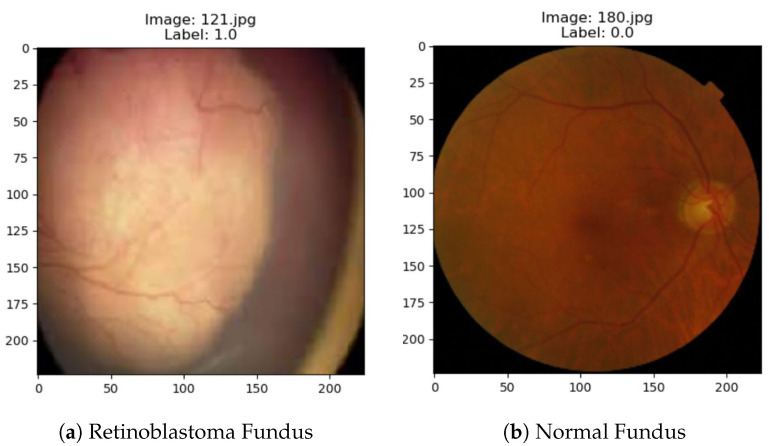
Sample image from dataset: (**a**) retinoblastoma fundus and (**b**) normal fundus.

**Figure 5 diagnostics-13-01932-f005:**
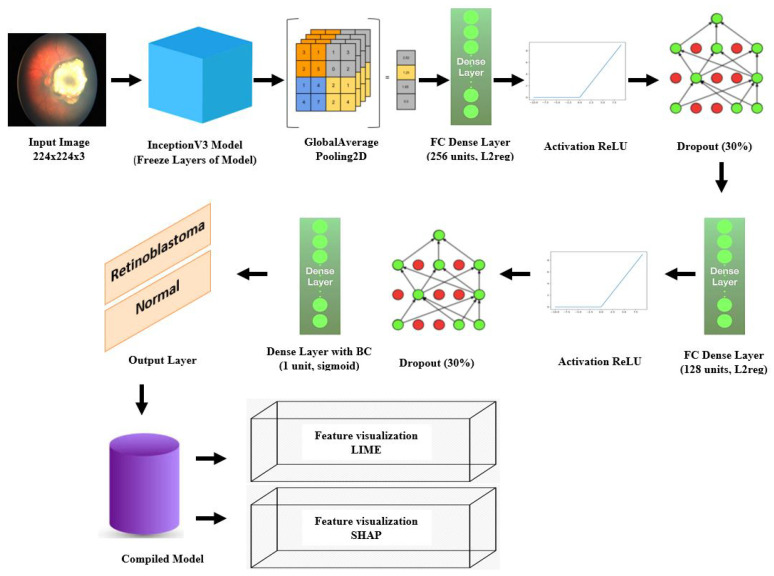
Architecture of transfer learning with InceptionV3 model for retinoblastoma classification, including LIME and SHAP visualizations.

**Figure 6 diagnostics-13-01932-f006:**
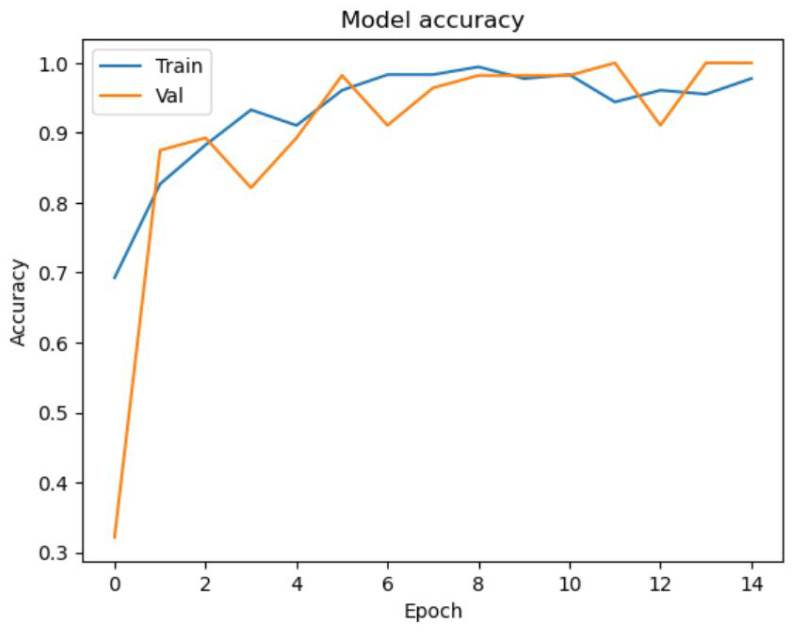
Accuracy Curve for Training and Validation for the Initial Epochs.

**Figure 7 diagnostics-13-01932-f007:**
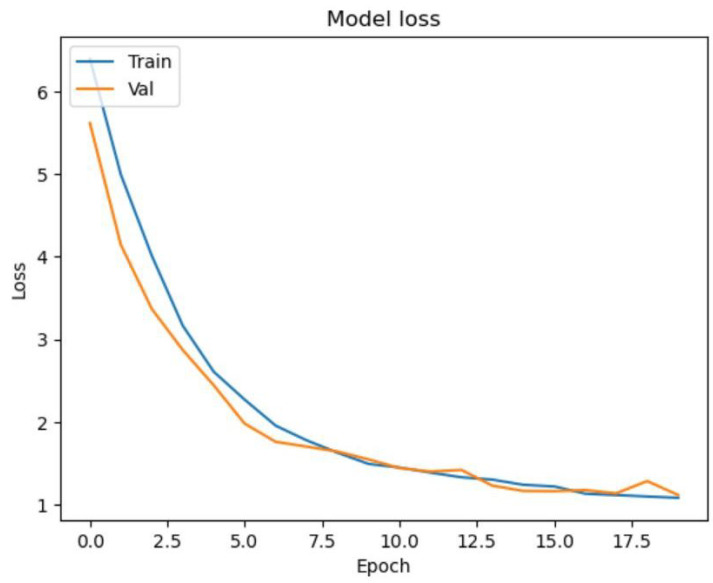
Loss Curve for Training and Validation for the Initial Epochs.

**Figure 8 diagnostics-13-01932-f008:**
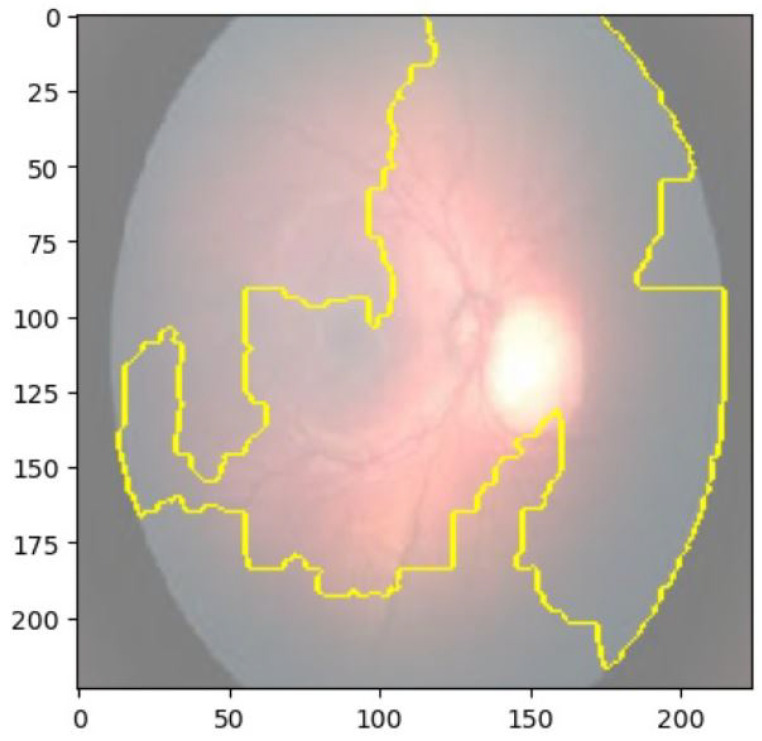
Saliency Map of Retinoblastoma Image 1 Using LIME.

**Figure 9 diagnostics-13-01932-f009:**
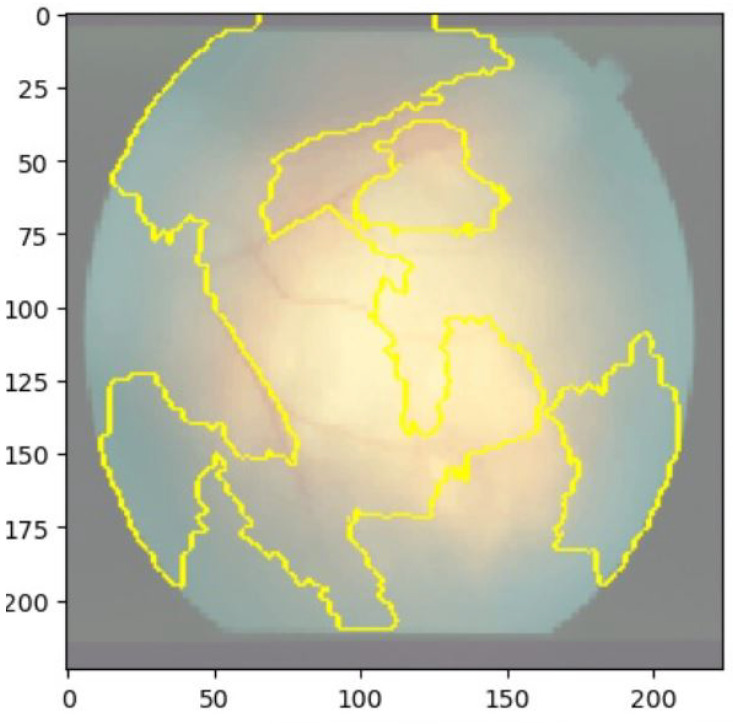
Saliency Map of Retinoblastoma Image 2 USING LIME.

**Figure 10 diagnostics-13-01932-f010:**
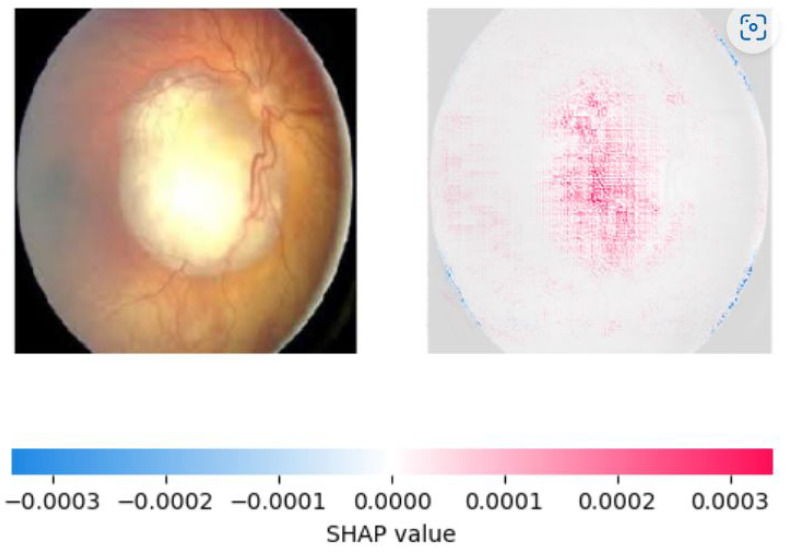
Saliency Map of Retinoblastoma Image 1 USING SHAP.

**Figure 11 diagnostics-13-01932-f011:**
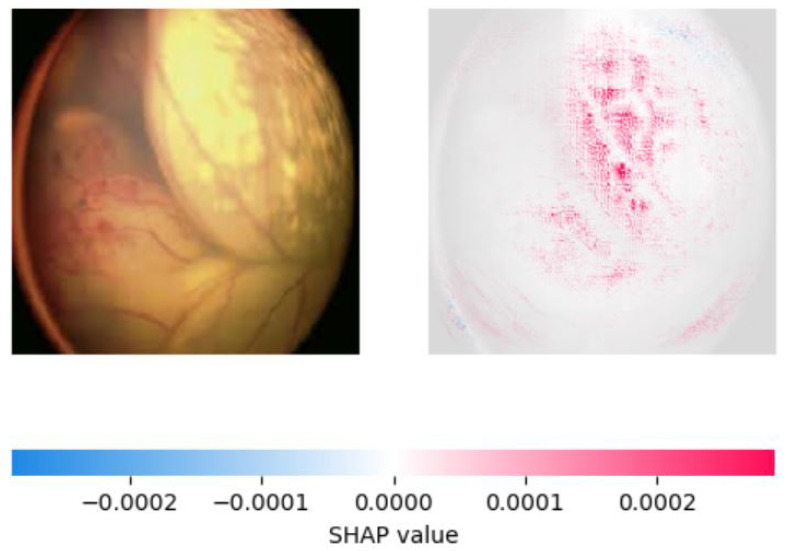
Saliency Map of Retinoblastoma Image 2 USING SHAP.

**Figure 12 diagnostics-13-01932-f012:**
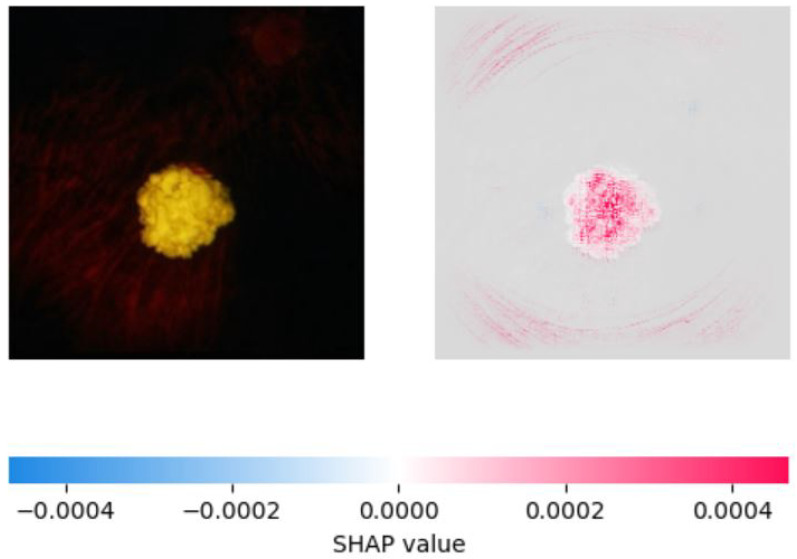
Saliency Map of Retinoblastoma Image 3 USING SHAP.

**Figure 13 diagnostics-13-01932-f013:**
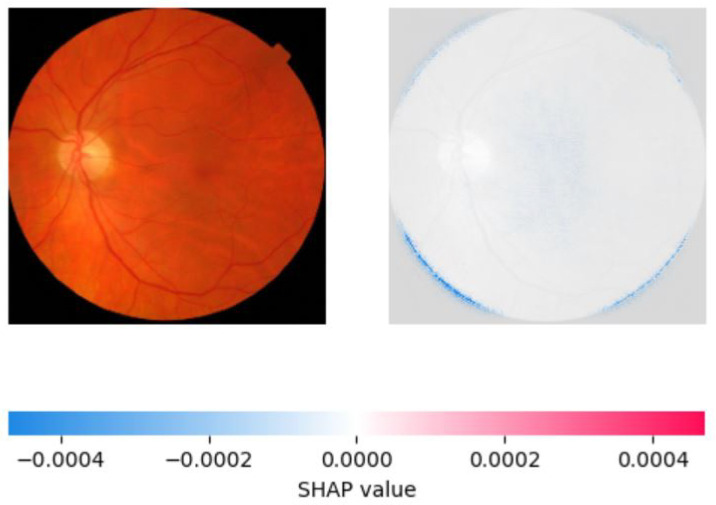
Saliency Map of Normal Fundus Image 1 USING SHAP.

**Figure 14 diagnostics-13-01932-f014:**
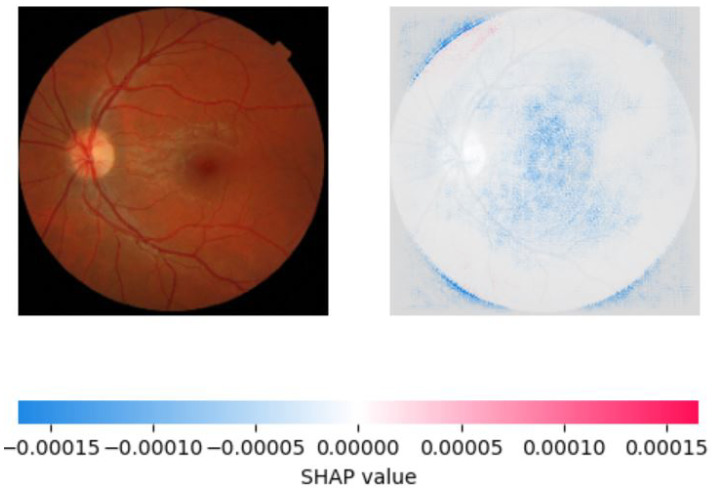
Saliency Map of Normal Fundus Image 2 USING SHAP.

**Table 1 diagnostics-13-01932-t001:** Optimized Hyperparameter Values for Model Training.

Hyperparameter	Value
Learning rate	0.001
Number of epochs	100
Batch size	32
Dense layer 1 size	256
Dense layer 1 reg	L2 (0.001)
Dropout 1	0.3
Dense layer 2 size	128
Dense layer 2 reg	L2 (0.001)
Dropout 1	0.3
Optimizer	Adam
Loss function	Binary crossentropy

**Table 2 diagnostics-13-01932-t002:** A Comparative Analysis of Our Model and Established Benchmarks.

Study	Model	Dataset	Performance	XAI Used
Kaliki et al. [52]	AI model	109 eyes, 771 fundus pictures	Sensitivity: 85%, Specificity: 99%, PPV: 99.6%, NPV: 67%	No
Kumar [54]	CNN-based ResNet50	Not specified	Accuracy: 93.16%	No
Jebaseeli et al. [55]	Linear predictive decision-based median filter, 2.75D CNN	Not specified	Accuracy: 98.84%, Sensitivity: 97.96%, Specificity: 98.32%	No
Rahdar et al. [56]	Semi-supervised segmentation method	80 fundus images	Not specified	No
Strijbis et al. [57]	Multi-view convolutional neural networks	40 retinoblastoma and 20 healthy eyes	Not specified	No
Henning et al. [58]	CNNs	Flickr training images	Low error rates (<3%)	No
Subrahmanyeswara et al. [59]	Deep Visual Geometry Group-net CNN classifier	Not specified	Not specified	No
Zhang et al. [60]	CNN(ResNet)	Not specified	Not specified	Feature Heatmaps Grad-CAM
Our Model	Transfer learning Inceptionv3	800 fundus images	Testing accuracy: 97%, Precision: 98.8%, Recall: 99.6%, F1 score: 99.2%	Feature Heatmaps Grad-CAM

## Data Availability

Not applicable.

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
