# Peer review of "Explainable AI for Retinoblastoma Diagnosis: Interpreting Deep Learning Models with LIME and SHAP"

_diagnostics, 2023, doi:10.3390/diagnostics13111932_

Round 1

Reviewer 1 Report

Retinoblastoma is a rare and aggressive form of childhood eye cancer that requires prompt 1 diagnosis and treatment to prevent vision loss and even death.  Deep learning models have shown 2 promising results in detecting retinoblastoma from fundus images. Improvement of the AI evaluation gives better sensitivity of algorythms in detecting this dangerous cancer. I consider this study to have valuable data that would be of interest if published. However it needs a major revision. The major concern is lack of materials and methods section i.e. the reaserch design and methodology must be improved. You should also discussed as potential limitation of this technology the lack of cheap techniques to take images of eye fundus in children.

Author Response

Reviewer 1

The major concern is lack of materials and methods section i.e. the research design and methodology must be improved.

Thank you for your valuable feedback on our manuscript. We sincerely appreciate your suggestions for improving the research design and methodology. We have carefully considered your comments and made the necessary revisions to address these concerns.

In the updated version of the manuscript, we have taken your suggestion into account and renamed the "Design" section as "Materials and Methods." We believe that this new title is more appropriate and accurately reflects the content of the section, providing a clearer understanding of the research methodology.

Furthermore, to enhance the clarity and visual representation of the research process, we have included Figure 3 in the manuscript. This figure illustrates the overall framework of our study, highlighting the sequential steps involved in data collection, preprocessing, model training, and the application of explainable AI techniques (LIME and SHAP) for generating interpretability and explanations.

The changes have been highlighted in the updated pdf.

You should also discussed as potential limitation of this technology the lack of cheap techniques to take images of eye fundus in children

Thank you for your insightful comment regarding the potential limitation of our proposed technology. We appreciate your suggestion to discuss the lack of affordable techniques for capturing images of the eye fundus in children. We have carefully considered this limitation and made the necessary additions to our manuscript.

In the revised version, we have included a dedicated section on limitations before the conclusion. Within this section, we specifically address the potential drawback you highlighted regarding the lack of cost-effective techniques for capturing eye fundus images in children

We believe that by discussing this limitation, we contribute to a more balanced and realistic evaluation of the potential impact and feasibility of our approach.

Reviewer 2 Report

In this study, Aldughayfiq et al. evaluated an AI model for the diagnosis of retinoblastoma. Below are my comments:

1. Given that this is a study of retinoblastoma, this manuscript needs to introduce/discuss the current approach to diagnosis of retinoblastoma, and how an AI model is considered to be helpful in that clinical context. Some countries have screening systems, other have opportunistic screening. All of these clinical aspects needs to be explained to better understand how this model provides any value. 

2. This study is based on a dataset + Google Images. Here, we have the textbook bias. All textbook cases are clearly distinguishable, whereas clinical practice can be difficult. Some cases may look like retinoblastoma, but be something else, or vice-versa. This bias should be discussed in detail as a study limitation.

3. This model is developed and trained on a certain dataset. But we have no idea of its performance in real-life. This is a very important limitation, which needs to be discussed and outlined very clearly.

Author Response

Reviewer 2

Given that this is a study of retinoblastoma, this manuscript needs to introduce/discuss the current approach to diagnosis of retinoblastoma, and how an AI model is considered to be helpful in that clinical context. Some countries have screening systems, other have opportunistic screening. All of these clinical aspects needs to be explained to better understand how this model provides any value.

Thank you for your valuable feedback regarding the introduction and discussion of the current approach to the diagnosis of retinoblastoma, as well as the clinical context in which an AI model can be beneficial. We appreciate your suggestion to provide a comprehensive understanding of these clinical aspects to better highlight the value of our model.

In response to your comments, we have thoroughly revised the literature review section of our manuscript. We have now included a detailed discussion on the current approaches to the diagnosis of retinoblastoma, considering various screening systems employed in different countries, as well as opportunistic screening methods. By incorporating this information, we aim to provide readers with a better understanding of the existing clinical practices and challenges associated with retinoblastoma diagnosis.

To ensure the visibility of these updates, we have highlighted the relevant text in the revised PDF of our manuscript.

This study is based on a dataset + Google Images. Here, we have the textbook bias. All textbook cases are clearly distinguishable, whereas clinical practice can be difficult. Some cases may look like retinoblastoma, but be something else, or vice-versa. This bias should be discussed in detail as a study limitation

We appreciate your insightful comment regarding the potential textbook bias in our study, and we agree that this bias should be addressed as a study limitation. We have carefully considered this concern and made sure to explicitly discuss it in the limitations section of our manuscript.

In the revised version, we have included a dedicated section on limitations before the conclusion.

We emphasized that while textbook cases often represent clear and well-defined instances of retinoblastoma, clinical practice can be more challenging. There can be cases that resemble retinoblastoma but are associated with other pathologies, as well as cases where retinoblastoma may be misinterpreted as something else. By acknowledging this limitation, we aim to provide a comprehensive assessment of the potential limitations of our study

Furthermore, we have highlighted this specific limitation in the revised PDF of our manuscript

This model is developed and trained on a certain dataset. But we have no idea of its performance in real-life. This is a very important limitation, which needs to be discussed and outlined very clearly.

Thank you for emphasizing the importance of discussing the performance of our model in real-life clinical settings as a significant limitation. We wholeheartedly agree with your point, and we have taken this limitation into consideration in our manuscript.

In response to your comment, we have provided a detailed discussion of this limitation in the dedicated limitations section of our revised manuscript. We have clearly outlined the need for further evaluation of our model's performance in real-life scenarios, considering various factors such as image quality, patient demographics, and clinical conditions

To ensure the visibility of this discussion, we have highlighted the corresponding text in the updated PDF of our manuscript

Round 2

Reviewer 1 Report

Now, the manuscript looks fine.